# How Do People Who Are Homeless Find Out about Local Health and Social Care Services: A Mixed Method Study

**DOI:** 10.3390/ijerph19010046

**Published:** 2021-12-21

**Authors:** Vanessa Heaslip, Sue Green, Bibha Simkhada, Huseyin Dogan, Stephen Richer

**Affiliations:** 1Department of Nursing Science, Faculty of Health and Social Sciences (FHSS), Bournemouth University, Bournemouth Gateway Building, St. Paul’s Lane, Bournemouth BH8 8GP, UK; vheaslip@bournemouth.ac.uk (V.H.); sgreen@bournemouth.ac.uk (S.G.); 2Department of Social Science, Stavanger University, 4021 Stavanger, Norway; 3Department of Nursing and Midwifery, University of Huddersfield, Huddersfield HD1 3DH, UK; b.d.simkhada@hud.ac.uk; 4Department of Computing and Informatics, Faculty of Science and Technology, Bournemouth University, Bournemouth BH8 8GP, UK; hdogan@bournemouth.ac.uk

**Keywords:** social exclusion, homelessness, technology, vulnerable, mental health

## Abstract

Background: There are significant numbers of people experiencing homelessness both in the UK and internationally. People who are homeless are much more likely to die prematurely and, therefore, need strong access to ongoing health and social care support if we hope to address the health disparity they face. Objectives: The aim of the research was to explore how people who are homeless identify and locate appropriate health and social care services. Design: A mixed methods research study was applied on people who are currently homeless or had previously experienced homelessness. Settings: The research study was based in an urban area in the southwest of England. The area was chosen as it was identified to be in the top 24 local authorities for the number of homeless individuals. Participants: A hundred individuals participated in the survey, of those 32% were living on the streets whilst 68% were living in temporary accommodation such as a charity home, shelter or a hotel paid for by the local authority. In addition, 16 participated in either a focus group or one-to-one interview Methods: The quantitative component consisted of a paper-based questionnaire whilst the qualitative aspect was focus groups/one-to-one interviews. The COREQ criteria were used in the report of the qualitative aspects of the study. Results: Quantitative data identified poor health in 90% of the sample. Access to both healthcare and wider wellbeing services (housing and food) was problematic and support for this was largely through third sector charity organisations. Qualitative data identified numerous systemic, individual and cultural obstacles, leaving difficulty for people in terms of knowing who to contact and how to access services, largely relying on word of mouth of other people who are homeless. Conclusions: In order to address health inequities experienced by people who are homeless, there is a need to review how information regarding local health and wider wellbeing services is provided in local communities.

## 1. Background

There is currently no internationally agreed definition of homelessness [1], with different countries identifying different criteria in their homelessness reporting. There is a European Typology of homelessness and housing exclusion which identifies different types of homelessness (Table 1); however, this is not universally used. Reasons for homelessness are multi-faceted, but fundamentally, at the core, it links to poverty, inequality and a lack of affordable housing; homelessness affects people of all ages and individuals from a range of economic, social and cultural backgrounds [2].

Due to this lack of agreed definition, it is hard to provide clear data pertaining to the numbers of homelessness internationally, although it is expected to be in excess of 2.1 million across 36 countries (OECD 2021) [1]. Certainly, in Europe and the USA, the situation seems bleak. Across the European Union, 24 out of 28 countries report that homelessness has increased significantly over the last decade [3]. In the United States, 580,000 people were experiencing homelessness at the beginning of 2020, which is an increase of 6% in 2 years. [4]. With the COVID-19 pandemic and associated economic hardship, this can only result in more long-term homelessness in the coming decade. Some studies in the EU [5,6] and the USA [7,8,9] have examined the issue of homelessness and health care access. All conclude that more can be realized with the utilization of technology and e-health applications. However, messages are mixed. Certainly, different systems, governments and cultures view homeless populations in different ways. It is, therefore, inevitable that the systemic difficulties and needs of a homeless population vary between cultures; consequently, a detailed and bespoke approach to the specific needs of a population is imperative.

Before the COVID-19 pandemic, homelessness across the UK was on the rise in that 1768 people were identified as homeless in 2010, rising to 4677 in 2018 [10]. The southwest has the third highest number of rough sleepers across England (Homeless Link 2020) [11]. This research was carried out in an area with a large homeless population, one of the highest in the UK [12]. During the COVID-19 pandemic, the numbers of those rough sleeping was reduced temporarily as local authorities were instructed to move those sleeping rough into temporary accommodation to enable self-isolation to occur, largely achieved using empty hotel accommodation.

Health outcomes for individuals who are homeless are poorer than that of the general population [13]. The mean age at death for males who are homeless is 46 years and for females 43 years compared to the means average age at death for people living in homes, which for males is 76 years and for females 81 years [14]. This pattern is also reflected internationally as a systematic review and meta-analysis across 38 countries by Aldridge et al. (2018) [15], who identified all-cause mortality and standardised mortality ratios of 7.9 for men and 11.9 for women. In addition to higher mortality, people who are homeless live with complex mental health [16] and physical issues [17]. Concerns regarding the health impact of homelessness resulted in recognition of the need for public health interventions to address health inequalities and disparities people who are homeless face. The NHS 5 Year Forward view [18] recognised the need for NHS to change their focus, recognising the need for a more engaged relationship with users to promote wellbeing and prevent ill health by breaking down barriers and supporting individuals in self-managing their own health. A review into the experiences of people living with complex needs (especially those who were rough sleeping) by Public Health England (2018) [5] concluded that interventions to promote good health in people who are homeless required multi-agency interventions such as pharmacology, psychosocial and disease prevention, as well as gender tailored interventions. It also recognised that no single intervention was effective but rather what was needed was a system wide integrated approach to meet the complex needs of these individuals.

In 2018, the Rough Sleeping Strategy was published [19] with a mission to eradicate rough sleeping by 2027. Integral to this is to support those already homeless and to provide more secure accommodation. As previously noted, the reasons leading to homelessness are complex; therefore, this requires homeless individuals to be able to access health/social care (such as GP, dentists, psychiatrists, psychologists and social workers) and wider wellbeing services (such as food and showers, etc.). The roles of health and social care professionals in this are crucial as health professionals can identify risk of poor health and put strategies into place to manage this, as well as promoting better health outcomes in those who have complex co-morbidities [20], whilst social care staff can support individuals in addressing financial and housing issues as some of the root causes of homelessness and support them in finding accommodation. However, there has been little work focused on exploring how homelessness individuals navigate and access these services, which is the topic this research aims to address. The aim of the research was to understand how people who are homeless identify and locate appropriate health and social care services.

## 2. Methods

### 2.1. Study Design

This study utilised a sequential mixed-methods approach [21], enabling methodological triangulation which McKim (2017) [22] argues results in more rigorous results. In addition, as homelessness is a complex social and human phenomenon, mixed methods enables us to understand these complexities more fully than would be possible from a single approach [23]. Data were collected using both quantitative and qualitative methods. COREQ reporting guidelines [24] were used in the framing and reporting of the qualitative aspect of this research.

### 2.2. Study Sample (Participants)

Recruitment from the study consisted of both purposeful and snowball sampling. People who were homeless across Poole, Bournemouth and Boscombe (Dorset, UK) were approached, and the purpose of the study was explained. If they indicated they were interested in participating, then they were given a participant information and participant agreement form. These were also read out as it was recognised that homeless people can have lower literacy levels [25]. Night shelters and supported accommodation were also purposefully approached. Latterly, snowballing occurred as participants identified other potential participants. In the main study, participants were approached face-to-face on the streets. In total, 100 participants participated in the questionnaire section of the study. All were either currently homeless (31) or had been recently rehoused in hotels/hostels (66) as a result of the COVID-19 pandemic. A very small number (3) were living in relatively long term supported accommodation and had been there for 18 months. Therefore, all the sample had had very recent experience of living in absolute homelessness. Furthermore, 16 participants were recruited for the qualitative interviews and focus groups (all of these focus-group/one-to-one interview participants had also completed the quantitative questionnaire). Estimates for the homeless population of Poole and Bournemouth vary, not unsurprisingly in a frequently itinerant population, and consequently any reports of study power would be estimated. However, a recent British Broadcasting Corporation report [26] estimated that 222 homeless people were rehoused in Poole and Bournemouth due to the COVID-19 pandemic, and based on this our study, power would represent a ‘good power size’ [27], pβ = 80%.

#### 2.2.1. Quantitative Data Collection

A paper-based questionnaire sought basic demographic information (date of birth, gender and whether they were currently or had previously slept on the streets), as well as a mixture of rating scale, multiple response and free-text questions related to the availability and accessibility of homeless services in the local area and their experiences of utilising mobile phones and the internet. As the primary purpose of the questionnaire was to identify access and utlisation of local services as well as internet usage, it was not possible to use a validated questionnaire. However, the questionnaire was developed by using a steering group, including a GP, mental health practitioners, academics and third sector organisations, all of whom had expertise in working with people who were homeless. In addition, the questionnaire was reviewed by a member of the steering group who had experience of homelessness and amended the questionnaire to ensure that the language used in the questionnaire was at an appropriate level. The questionnaire also sought information on participant’s perceived health and wellbeing; however, in order to limit the size of the questionnaire it was not possible to include validated scales. Cohudhury et al. [28] argues it is important to keep questionnaires short when working with vulnerable populations.

Questionnaires were conducted by one member of the research team and were conducted either on the street or in cafes. Support was given to those who struggled with literacy, and on occasion questions and response options were read out and clarified by the researcher. At the end of the questionnaire, participants were asked if they would be interested in participating in a focus group/one-to-one interview and recruited accordingly.

Table 2 shows age and gender characteristics of participants for the qualitative questionnaire. A total sample of 100 individuals who were homeless or previously homeless were recruited into the study. The age range of the studied sample was diverse (51 years). Seventy-one percent of participants were male, and 29% were female; this is significantly higher than the UK homeless population, which is approximately 14% female [10].

#### 2.2.2. Qualitative Data Collection

Participants for the qualitative arm of the study varied in age between 22 and 66, with a mean of 41.85 (7.12). Of those, 75% were male, and 25% were female. The length of time experiencing homeless varied from 4 week to 20 years, and half of the participants were currently homeless, with the rest living in supported accommodation (see Table 3). All had experience, past or present, of homelessness.

Qualitative data collection consisted of one-to-one interview/focus groups. Focus groups were chosen as it enabled us to proceed beyond the data obtained in the questionnaire, providing a greater depth of understanding regarding the participant’s experiences [29]. However, the focus groups were found to be challenging as many of the participants had ongoing addiction issues (drugs and/or alcohol), and as such focus and concentration in a group setting were more challenging [30]. Due to this, we decided to conduct one-to-one interviews in order to better enable the participants to engage. Qualitative data collection was undertaken by two members of research team who have extensive experience of working with marginalised groups and undertaking qualitative research. Data collection occurred during the COVID-19 pandemic when restrictions in the UK were relaxed, and as such focus groups and one-to-one interviews were conducted outside on church grounds, supported accommodation gardens and places that were quiet; huts, the participants felt comfortable in sharing their experiences. All interviews/focus groups were audio recorded for later transcription, enabling the researchers to focus on what the participants said. An interview guide (Table 4) was used in both focus groups and one-to-one interview to ensure consistency in the interview process. Focus groups lasted approximately 45 min whilst and one-to-one interviews lasted approximately 15 min. Interviews were relatively short due to poor concentration amongst all participants. Of the 16 participants involved in the focus groups and 1:1 interviews, all but one had a current or recent drug/alcohol addiction. Furthermore, 9 had current mental health diagnoses. Participants found focus and concentration difficult; consequently, shorter interviews were considered preferable as a means to gather information in an effective manner [31]. Indeed, in a wide-ranging review of research with vulnerable populations [31], face-to-face qualitative interviews were found to be an effective strategy for overcoming cognitive difficulties and issues of mistrust.

### 2.3. Ethical Considerations

Ethical approval was obtained from the Bournemouth University Research Ethics Committee. All participants were provided with a Participant Information Sheet and Agreement Form and consented to participating. It was made clear to participants the voluntary nature of the study and that their anonymity would be assured. As previously stated, researchers were sensitive to potential literacy and cognitive difficulties in the homeless population and offered assistance where needed to make the process as inclusive as possible. Individuals who participated were reimbursed for their time with a gift voucher for a local supermarket. As data collection occurred during the COVID-19 pandemic, we ensured that governmental guidance regarding social distancing was followed, and the majority of the data collection occurred outside. As such, we were mindful of the need for confidentiality and ensured that participants were happy sharing their stories where they were or moved to quieter areas when the environment was busy.

### 2.4. Data Analysis

Quantitative data were analysed using Statistical Package for Social Sciences (SPSS) software (IBM, version 27) [32]. The dataset was checked for outliers and missing replies. The amount of data missing from the completed questionnaires was negligible (less than 3%); thus, missing items were replaced with the sample mean [33]. Quantitative analysis was mainly limited to descriptive statistics. More in-depth analysis was carried out using one-way ANOVAs and *t*-tests to examine differences across gender and age for technology ownership/accessibility and user ability.

Qualitative data from focus groups/one-to-one interview were audio recorded and transcribed verbatim. Transcribed accounts were analysed by two members of research team using inductive thematic analysis [34]. The first stage of the analysis included reading each individual transcript, highlighting key words or phrases to identify initial codes. These initial codes were then analysed to identify potential themes. Potential themes were then considered across the entire dataset, resulting in the development of a thematic map which was shared with the rest of the research team to ensure credibility of the analytical process. Transcripts were not sent back to participants for member checking for two reasons: firstly, pragmatically due to the literacy challenges for people who are homeless; and, secondly, Thomas (2017) [35] argues that there is no evidence that member checking enhances the credibility or trustworthiness of qualitative data.

## 3. Results

### 3.1. Quantitative Data

#### 3.1.1. General Health

Poor health was highlighted by most participants, with ongoing health issues present amongst 90% of participants. Although varied, the main issues were mental illness (72%) and addiction (69%) (Figure 1). Furthermore, self-reported reading ability was found to be poor across the population as 30% of participants reported low confidence in their reading ability and almost one-fifth (18%) reported no confidence at all. Approximately 25% of the participants required some assistance when completing the questionnaire. There was a gender difference in self-reported reading confidence with 72 male participants reporting greater confidence (M = 3.12, SD = 1.41) than the 28 female participants (M = 2.75, SD = 1.40), and this was found to be statistically significant, t(97) = 3.087, *p* = 0.003.

Two-thirds of participants were on regular medication. Of these, 36% of participants on medication were for mental health issues and 25% were for addiction difficulties (mainly methadone). The remaining medications were for cardiac (beta-blockers and blood thinners) and respiratory (inhalers) complaints. Issues with prescriptions and the collection of medication were prevent amongst participants. Although stock shortages and the COVID-19 crisis accounted for 38% of the issues highlighted by participants, the main cause of medication problems (62%) was that of forgetting either to obtain a repeat prescription or to pick up the medication.

There were also notable gender differences within the general health of the participants. Analysis found that male participants (*n* = 72) were less likely to have mental illness (M = 0.61, SD = 0.491) than the female (*n* = 28) participants, (M = 0.860, SD = 0.356) and this was found to be statistically significant, t(95) = −2.413, *p* = 0.018. Furthermore, male (*n* = 72) participants were less likely to have substance addiction problems (M = 0.51, SD = 0.503) than female (*n* = 29) participants (M = 0.860, SD = 0.536), and this was found to be statistically significant, t(95) = −3.297, *p* = 0.001.

#### 3.1.2. Healthcare Access

A variety of professional health and wellbeing services were accessed by participants on a regular basis. Local homeless charities were frequently used to access doctors and nurses via drop-in clinics for consultations, and around 60% of participants had used a local service in this fashion at least once in last 12 months. However, the first-place participants were likely to access medical care by accident and emergency, which was used regularly (over 10 times annually by 15% of participants). This was considered the most reliable method of receiving professional medical care.

Problems accessing health services were prevalent, with 47% of participants saying it was very difficult/difficult to access health care professionals, whilst 21% finding it relatively easy. However, it was those who were currently sleeping rough who had the most difficulty in accessing health services, with 62% of those currently sleeping rough saying that it was very difficult/difficult to access health care professionals compared to 49% of those who, in some form, had supported accommodation.

Furthermore, gender differences were prevalent amongst the homeless population regarding healthcare accessibility. Male (*n* = 65) participants found it easier to access health related services (M = 2.31, SD = 1.40) than compared to female (*n* = 25) participants (M = 1.88, SD = 0.98), and this was found to be statistically significant, t(90) = 1.74, *p* = 0.042. However, although males found it easier to access health services, they were *not* more likely to utilise them. In fact, male participants were *less* likely to access metal health services (M = 1.15, SD = 0.444) than compared to the 24 female participants (M = 1.38, SD = 0.647), and this was found to be statistically significant, t(85) = −1.829, *p* = 0.036. The same was found with addiction services, with male participants less likely to access addiction services (M = 1.31, SD = 0.593) than compared to female participants (M = 1.68, SD = 0.863), and this was again found to be statistically significant, *t*(85) = −2.337, *p* = 0.022.

Locating dental care and accessing a general practitioner (GP) were highlighted as particularly difficult. Participants highlighted a variety of reasons for these difficulties, including both systemic and personal reasons. A lack of permanent address (15%), long waiting times (12%) and poor signposting (20%) were amongst the systemic issues. On a personal level, participants highlighted anxiety (18%) and addiction problems (10%) as contributory factors to their difficulty accessing healthcare. A further 10% referred to prejudiced attitudes towards the homeless population as a barrier to healthcare access. Again, there was a difference between genders with male (*n* = 63) participants *less* likely to access dental services (M = 1.13, SD = 0.421) than compared to female (*n* = 24) participants (M = 1.25, SD = 0.090), although this was not found to be statistically significant, t(85) = −1.202, *p* = 0.233.

#### 3.1.3. Wellbeing Management

With regards to accessing services for general care management (food, keeping dry, finding shelter and financial support), local charities and churches were the main sources of assistance. Forty percent of participants had used soup kitchens and foodbanks at least 10 times in the last 12 months. The YMCA (25%), BH1, a local charity (26%) and another local charity (21%) had all been utilised 10 or more times in last 12 months. However, statutory care services such as social workers, housing officers, job centres and the town hall were all rarely used by participants, and less than 6% used these services regularly.

By exploring how homeless people identified these local services, participants highlighted that word of mouth (22%), support worker advice (18%), volunteer advice (18%) and local knowledge (16%) as pertinent factors. However, across the survey, 41% said it was difficult or very difficult to know how to access services, and only 19% found it relatively easy. Again, there was a discrepancy between those who are currently homeless and those in supported accommodation. Forty-nine percent of those currently sleeping rough said it was very difficult/difficult to know how to access services compared to 33% of those who are, in some form, receiving supported accommodation. Reasons for these difficulties were varied and systemic and personal in nature. Having no permanent address (18%) made accessing some services difficult, as did knowing where to go (14%), prohibitive distances (14%) and having no internet access (17%). Anxiety (12%) and low mood (14%) were personal reasons that made finding and accessing these services difficult. Gender differences were also apparent, with male participants (*n* = 72) more likely to know where to access basic needs (shelter, food, clothing, showers, etc.) (M = 2.80, SD = 1.093) than compared to female (*n* = 24) participants (M = 2.29, SD = 0.999), and this was found to be statistically significant, t(98) = 1.991, *p* = 0.050.

#### 3.1.4. Internet and Mobile Technology

Phone ownership amongst participants was high, with over three-quarters (77%) owning either a mobile or smartphone, whilst 23% had neither. A quarter owned a smartphone (Figure 2). Although phone ownership was high, only 50% of those asked could access the internet regularly. With difficulty accessing the internet due to a lack of data plan and Wi-Fi, participants tended to use the library (38%), internet cafes (36%), the Health Bus (16%) and night shelters (20%) to access an online connection. Male participants (*n* = 72) were more likely to be able to access the internet (M = 0.600, SD = 0.494) than the female (*n* = 28) participants (M = 0.250, SD = 0.441), and this was found to be statistically significant, t(97) = 3.249, *p* = 0.002.

Confidence using the internet is generally relatively poor amongst participants, with approximately half (49%) stating that they had low/very low confidence. Only 29% expressed high or very high competence (Figure 2). However, among the participants, internet confidence score and age were negatively correlated, r(97) = −0.178, *p* = 0.040. Older age was found to be a significant predictor of less confidence. Furthermore, male participants (*n* = 71) were more confident (M = 2.94, SD = 1.17) than female (*n* − 26) participants (M = 1.94, SD = 0.98), and this was found to be statistically significant, t(95) = 3.274, *p* = 0.001.

Participants were asked how easy it was to access the internet whilst living on the streets. Fifty-three percent stated that they found this difficult or very difficult, and only 9% said they found it easy/very easy. A lack of connection, data plans and Wi-Fi hotspots was highlighted. Battery life and charging was another problem highlighted by participants, and this is consistent with other research [36]. Twenty-five percent stated that they could only charge their phone and connect to the internet for a few hours a week. Of those asked, only 15% were able to maintain a phone and an internet connection consistently. Again, male (*n* = 67) participants who responded found it easier to access the internet (M = 2.31, SD = 1.40) than compared to female (*n* = 25) participants (M = 1.88, SD = 0.98), and this was found to be statistically significant, t(90) = 1.74, *p* = 0.042.

#### 3.1.5. Open Access and Information

Finally, participants were asked what services they would most want information on, if they had an open access technological device. A variety of services were highlighted; the most sought-after services included food banks (78%), benefit information (64%), soup kitchen location (62%) and addiction services (58%). Other services highlighted as desirable were those that offered shelter, support, a social outlet and general health and hygiene services.

### 3.2. Qualitative Data

Analysing data from the focus groups and one-to-one interview collated resulted in the identification of three overarching themes: systemic, individual and cultural obstacles (to accessing health and wellbeing information); the current means by which these barriers are overcome; and the need for intervention and assistance.

#### 3.2.1. Theme 1: Obstacles

##### Structural

Across the focus groups and 1:1 interviews, there was a strong prevailing sense of obstacles and barriers with respect to accessing services and accessing help for those who are homeless, and many of these are rooted in systemic, individual and cultural issues. Structural/systemic issues were highlighted by many participants, with a strong emphasis placed on a need for a ‘local connection’.

‘*When I moved down here it was quite hard for me at first because I had to have a local connection before I could sign up for anything*’.(P3, female, focus group)

In order to access services, such as housing support, one requirement of most local authorities is the notion of a local ‘connection’, for an individual to have been born in the area. In a frequently itinerant population, this can represent a huge barrier to accessing much needed services, especially for those with drug addictions as they choose to move to a new area to move away from their previous drug dealers and drug community. In order to compound the problem, some local authorities have an ‘expiry date’ on local connection; thus, a homeless individual returning to the town of their birth after a prolonged period may no longer be considered ‘local’. Again, this can compound and further complicate the situation.

A frequently itinerant population found a lack of national standardisation of housing systems, GPs and dentist identification requirements and issues with what constitutes a local connection added further confusion and difficulties for the homeless population and can have a direct negative impact on their health and wellbeing. Repeat prescriptions can be missed, in addition to follow-ups from mental health/drug alcohol appointments. Without a permanent address, it is very difficult to register with a GP, and those moving to a new area found this particularly difficult. Indeed, the need for a permanent address and identification was a source of difficulty highlighted consistently by participants.

‘*I couldn’t register with a GP until I had a permanent address*’.(P3, male, focus group)

‘*A major problem yeah like Photo ID, they all want Photo ID if you ain’t got Photo ID you can’t do anything and that’s a problem you go in there with like a dole office letter or a housing statement from this place or something like that but it’s not good enough they want Photo ID*’.(P12, male, interview)

This lack of ID often resulted in financial vulnerability. Numerous participants highlighted their inability to obtain a bank account due to a lack of ID and not having a permanent address. This resulted in them using someone else’s bank details to obtain their paid benefits, and this results in a subsequent risk of financial abuse, theft and manipulation.

‘*I got robbed because I had my money put into someone else’s account because I didn’t have a bank account and then they robbed me of £450′*.(P12, male, interview)

The protocols used by supporting accommodation services and night shelters were also highlighted as structural barriers to accessing homeless services. Often, individuals are required to ‘prove’ their homeless status, and this can be difficult and complex. Services will send outreach workers to areas where homeless people are known to sleep overnight and assess suitability and status accordingly. Often it was a requirement that the person had to be at the same place for three consecutive nights or else they lost their claim for a place in the night shelter and the process had to be started all over again. This is, at best, impractical and, at worst, dangerous. Participants were often frightened to return to the same place three nights running and felt this was making them more vulnerable.

‘*A local night shelter took me in but this time it took them 3 months just to verify me… had to find me in same place 3 days running and I don’t want to be there after first night*’.(P6, male, focus group)

The ubiquitous difficulties associated with prejudice were highlighted by the vast majority of participants as having an impact on their ability to access and utilise services. Many felt that the local council was particularly unhelpful with regards to the homeless population.

‘*It’s a tourist town innit, we are a nuisance and holidaymakers don’t like it. Council want us out of sight, in another town if possible…*’.(P7, male, interview)

Furthermore, participants felt that their addiction problems were treated with prejudice by health services such as GP and dentist, housing services and some night shelters, and the chances of finding accommodation were severely prejudiced as a result. Many also highlighted a lack of trust in many services considering this prejudice and found it difficult to approach some of these for assistance as a result.

‘*I feel like I’m being judged by them, even before I say anything…*’.(P15, male, interview)

‘*There’s just days where they just don’t wanna help you know what I mean and you get treated differently by staff if you’re homeless, in hospitals, at docs, on the streets…*’.(P7, male, interview)

A poorly organised system compounded by perceptions of prejudicial attitudes compounds the already weighty problems associated with homelessness. These systemic problems are further complicated by the unique personal circumstances of the homeless population.

##### Personal

Personal issues and circumstances were also found to present obstacles and barriers to accessing services, healthcare and other help for homeless individuals. Addiction problems were highlighted by many of the participants in making access to services much more difficult. Many of the shelters were highlighted as having prohibitive drug and alcohol policies. Emphasis was placed by participants on the difficulty in obtaining placement if struggling with addiction problems and the added risk of eviction from such placements in the case of relapse.

‘*Addiction got in the way of everything… night shelters, supported accommodation, everything*’.(P7, male, interview)

‘*They say about the-the health professionals you know about them not being judgemental and-and not-not judging anyone but-but God you know so many of them do so again and again, it just puts you off puts you off*’.(P14, male, interview)

Participants also highlighted the problems associated with accessing heath care services due to a lack of internet accessibility and competence in using online services. Much of the information available to homeless people is available online, but this is not always useful for the homeless population. The ubiquitous problem of the ‘digital divide’ has been somewhat overlooked by many services. Participants noted the difficulty in obtaining information on homeless services due to their personal circumstances. Particularly pertinent was the difficulty associated with applying for, monitoring and updating financial benefit claims. Others highlighted a lack of training/reading ability in the use of internet-based services and information. Even when they had overcome the barrier of internet access, many, particularly older generations, found using these online services difficult.

‘*We all need to an have internet connection in order to sort your benefits sorted out. Without internet it is very hard*’.(P16, female, interview)

Practical issues surround internet usage were also highlighted. These included the general expense of buying a phone and maintaining a data plan. Furthermore, participants highlighted struggles in keeping a phone charged, finding Wi-Fi hotspots and generally maintaining any online presence. Finally, theft was highlighted as concern. Some participants felt very vulnerable owning something that may make them a target for theft.

Participants highlighted traveling as problematic considering their personal circumstances. Accessing services was compounded by the lack of bespoke local information. Often, desired services were identified some distance away, and this caused problems for many participants. Those living and sleeping rough were particularly impacted. In addition to the expense of public transport, participants said that they were very reluctant to leave all their possessions to travel and taking everything with them was impractical.

‘*Then again it’s travel and stuff isn’t it and-and obviously it’s like prioritising things… sometimes it’s too far and I can’t leave my stuff or carry it all, I miss chances for supported shelter*’.(P7, male, interview)

Finally, participants highlighted female-specific barriers to accessing services and help for the homeless. One of the most pertinent issues was that of safety, with the issue of greater vulnerability for the female homeless population rarely addressed. Personal hygiene was also noted as a particular problem in the female population with showering facilities at a shelter that were often not private and rarely female specific. Female participants highlighted a difficulty in acquiring sanitary products and a general lack of privacy, which resulted in an element of embarrassment and in some cases a feeling of increased danger and vulnerability.

#### 3.2.2. How Barriers Are Overcome

The homeless population have consequently had to find coping strategies to ameliorate, at least partially, some of the impact of the challenges they face. Acquiring information about help is a high priority for the homeless population. The solutions related to finding places to eat, finding shelter, receiving medical advice and a myriad of other health and support services are relatively simple for the general population but less so for the homeless, particularly if regularly moving from town to town. Participants were unanimous in their experiences, stating that the best method to acquire information about services was by ‘word of mouth’.

‘*For soup kitchens and things, its word on the street, aye word on the street, the other homeless people*’.(P5, male, interview)

Most participants saw this as a positive thing and a useful method for finding services. However, some highlighted difficulties with such information. Firstly, it was not always reliable; more concerningly, some people had experiences of being sent to unsafe places when asking for information. Moreover, participants highlighted the fact that this information should be available 24 h a day from a reliable and up-to-date source (government or local authority).

With regards to the services available to help the homeless population, most participants identified the third sector, particularly the work of church-based charities and other homeless charities, as being a lifeline for them and often the very first port of call when finding oneself homeless. In addition to offering food and shelter, they also offered an outlet for social activities and meeting people. Furthermore, a doctor or nurse would run a drop-in clinic at these locations, which participants found useful, particularly considering the difficulty in registering with a GP.

‘*Everyone knows, the first thing you do is find a church, go there and ask for help… they save lives*’.(P5, male, interview)

‘*Government wise there is no (support) but the support is out there, there’s more support down here through churches places like BH1 and stuff like St Mungo’s, Half Time that you I’ve never seen that support anywhere*’.(P3, female, focus group)

Although participants specifically identified charities, churches and peers as ameliorating some of the impact of being homeless, they remained less positive about local authority and government funded services. There was a sense of distrust expressed by most participants towards the council, government-run services in general and an unreliable benefits system. However, the general feeling amongst participants towards the local police force was positive and most found them helpful and a useful source of information.

‘*The police are more tolerant than the local authorities. As long as you’re not causing any trouble or disturbing anyone you’re fine*’.(P2, female, focus group)

Within the focus groups and 1:1 interviews, it was clear that time spent on the street had an impact on how barriers were overcome. Those that had been homeless for a long time talked more of word of mouth, being street wise and knowing the system. Those who were first-time homeless or relatively new to homelessness tended to look to established charities and local authorities for advice and help. Participant (P7) below had been living on the streets for 9 years, contrasted with following participant (P14) who was having their first experience of homelessness. This disparity between ‘experienced’ homeless individuals and those relatively new to homelessness was apparent throughout the focus groups and interviews.

‘*It’s who you know iinit? The people on the steet know the score, know where its safe and where the food is*’.(P7 male, interview)

‘*The churches have helped a lot and the YMCA*’.(P13, female, interview)

#### 3.2.3. Theme 3: Intervention and Assistance

Participants felt that although there were many services offering support for homeless individuals in the local area, they were not signposted well enough.

‘*It’s there, but unless you know where things are, it’s kinda pointless*’.(P7, male, interview)

Participants felt that information regarding food, shelter and clothing/hygiene were the most important. These basic survival and self-care services were expressed as the highest priority by the vast majority of participants and, consequently, should be included in any technological application. These were considered ‘emergency’ services by many of the participants. Then, the secondary focus should be on finance (benefits), social support/housing and employment information. Participants also emphasised a need for up-to-date information, phone numbers, addresses and emails of these services, as they were liable to regular change and often out of date. They felt that technological application could be updated regularly and help ameliorate such difficulties.

‘*Up-to date info is needed, so I was looking at their notice board (at night shelter) and their list of soup kitchens was 4 years out of date alright so you know the other thing is support that you know this list must be sort of updated*’.(P1, male, focus group)

Again, different needs were apparent across age, gender and length of time being homeless. Firstly, age was often a predictor of how information was viewed. Older members of the homeless community tended to see a need for simplified information and require help with benefit claims and other online services.

This contrasted with younger homeless participants who wanted more information rather than help for simply accessing information. The emphasis for the younger homeless individuals lay within quality and richness of information; with older individuals, it was about help access any basic information and online services.

‘*Everthing is online now, if you want to make a claim you need a computer. A computer that we could use would help*’.(P11, male 66, interview)

‘*We need information on jobs, and more information on chemist opening times for prescriptions*’.(P7, male 39, interview)

Gender also influenced how the homeless population felt about what interventions and assistance would be desirable in the future. The emphasis for female participants was more likely to be safety and issues surrounding vulnerability and privacy.

‘*It would be good to know where is safe, which carparks to sleep in… some are dangerous and have druggies… some are okay*’.(P13, female, interview)

‘*Having a shower was quite hard even though there was open showers there, its easier for a man to have a shower there than a woman*’.(P3, female, focus group)

Finally, time spent being homeless appeared to influence participants’ view of what kind of information would be useful when living on the streets. Those who had been on the streets for a long time identified a need for information relating to basics such as food, shelter and money. Those who were relatively new to being homeless were more likely to want information relating to opportunity, jobs, housing, etc. Participants who were relatively new to the streets also identified a need for being clean, presentable and not ‘appearing homeless’.

‘*I like to keep myself clean and presentable that’s the main thing of-of-of presenting yourself as like someone who doesn’t look homeless you know so if you act normal look like the others then, if you’re clean shaven*’.(P8, male, interview, 3 months homeless)

## 4. Discussion

The objective of the study was to investigate what services were available in the local area for the homeless population, how easily these services were to access, what made access more difficult and what is being/could be performed to ameliorate any identified difficulties. Both qualitative and quantitative data support various findings.

Difficulties in accessing services were highlighted extensively across qualitative data; as a result of interviews/focus group discussion, various barriers were identified, both systemic and personal. It was evident that current infrastructure worked against the homeless population. Particular emphasis was placed on the difficulty faced by the homeless population regarding both the need for a local connection and the issue of needing accepted photographic identification to access many services. This notion of a local connection is vital as it is on this basis that an individual can access assistance from the local authority. Local connection is defined [37] as follows:(a)They are, or were in the past, normally a resident there for at least 6 months during the past 12 months or for 3 years during the past 5 years, and that residence was of their own choice.(b)They are employed there.(c)They have family associations living in the area for at least 5 years.(d)Of any special circumstances, such as the need to be near special medical or support services, which are available only in a particular district.

For those who were homeless that were living with drug addiction, moving away from the local area to somewhere new was often seen as a method to try to say clean, as they moved away from their drug associates. However, this made it more challenging for individuals to be able to access support due to a lack of a local connection as they had to be resident in that area for 6 months before they could access help. Moreover, the challenge was that they had to have to evidence of being in that area for six months; however, how is that achieved if one is living on the streets? Furthermore, findings suggest that time spent on the street has an impact on the type of services homeless people attempt to access. Newer homeless individuals tend to look for housing support and employment support, which is a method of escaping their situation. That contrasts to the long-term homeless individuals who focus primarily on basics, food, warmth and money, with less of a view of escaping their situation and perhaps a sense of resignation.

The lack of signposting for services was identified by two-thirds of the participants in the questionnaire and was a pertinent theme amongst focus groups/one-to-one interviews. The homeless population simply did not have access to relevant, up-to-date and useful information. Even though services are available, a systemic failure of accurate, focused communication compounds an already difficult situation, particularly for those new to the locality. There were age differences identified in the qualitative data as to the type of information and signposting desired. Younger people tended to focus on the quality of the information, whereas older people were more concerned with how to access any information. Perceived institutional prejudice was also identified, and participants highlighted a feeling that they found it harder to access services such as GP and dentist due to their homeless status. This results in many people who are homeless having to access emergency departments when experiencing a crisis in care, resulting in accessing an emergency department for people who are homeless in the last 7 years [38]. This is problematic as it reduces access to both preventative and ongoing care through primary care services, reducing opportunities for health promotion as well as perpetuating and compounding both the management of pre-existing complex chronic illness as well higher mortality perpetuating the health disparity experienced by this group.

Personal circumstances were identified across both the questionnaire and the focus groups/interviews as detrimental to the identification and access of services. General health was found to be poor with over 50% of those questioned identifying as having mental illness and/or addiction issues. This was further highlighted in focus groups and interviews where participants highlighted addiction as a barrier to identifying and utilising services. Gender differences were prevalent regarding mental health and addiction. Female participants were significantly more likely to have mental illness and significantly more likely to have addiction problems. Although the mental health findings are supported in the other literature [39], addiction differences across gender were less expected and further research would be beneficial in this area.

Education and reading confidence were also personal barriers to accessing services, and 30% of participants reported low confidence in their reading ability. Gender differences were also apparent, with female participants being significantly less confident when self-reporting their reading ability. Poor literacy was highlighted during interviews with participants feeling that a significant minority of homeless people were being overlooked due to their reading ability. This barrier was further compounded by a lack of confidence when using the internet. In light of the rapid switch to online based services (including benefit claims), it was of concern that 49% of participants had low/very low confidence in internet use. With the additional practical issues of internet use for the homeless, only 50% could access the internet regularly, and with the connectivity and charging issues highlighted during interviews, it is of little surprise that this ‘digital divide’ compounds the difficulties faced by the homeless population. Despite this, participants indicated a potential going-forward for technology-based health advice and information sharing. Participants were very supportive of a publicly accessible technological application for obtaining information bespoke to a local area. Positive views were expressed about a system that could be regularly updated to provide information that was relevant, useful and current. Participants felt that priority should be given to information about food banks, benefit information, soup kitchen location and addiction services. Doubts about charging, connectivity and cost of mobile devices were still expressed amongst many participants; however, there was strong support for a more static approach. Digital technology has the potential to transform health care enabling different ways for people to be able to access health information as well as use technology to self-manage their ongoing health care [40]. Recent projects [41] for addressing this have looked at increasing technology in charities working with the homeless to promote their digital inclusion and confidence in accessing m-health technology, and whilst these appear to be improving confidence, the health impacts from these projects appear not to have been formally evaluated.

The difficulties faced by the homeless population were found to be ameliorated to some extent in a variety of ways. ‘Word-of-mouth’ played a large role for many of those interviewed and those in the focus groups. There seems to be a heavy reliance on ‘the word on the street’, and participants highlighted a strong sense of community when needing information. Time spent living on the street had a significant bearing on word-of-mouth utilization. Those who had spent more time homeless were significantly more likely to use word-of-mouth for information gathering, whilst people new to homelessness were more likely to approach social services and established systems. Some of the longer-term homeless participants found themselves at the hub of information sharing and were a vital link between services and those on the streets (particularly those who were from out of area or first time homeless). Although this is mainly viewed as positive, some participants also highlighted the inherent dangers of such information sharing, including the risk of misinformation and potential exploitation from less trustworthy sources. However, in the absence of accurate and easily obtained information, this source of information is often the only option. Third party services (charities, churches and individual helpers) were found to be a lifeline to a large percentage of the homeless community and were considered invaluable. Soup kitchens, YMCA and two local charities were visited with the highest frequency by those who completed the questionnaire. During the focus groups/interviews, it was clear that without these services, life would be considerably more difficult for the homeless population. When seeking advice, it was again the charity sector that was most approachable according to participants. Quantitative data indicated that 68% of participants would receive advice about available services from charitable organisations, with very few seeking professional services such as the council. This was confirmed within the focus groups, who rarely sought information from official services due to a combination of mistrust, long waiting times and poor previous experiences.

As previously noted, both Public Health England (2018) [6] and the Ministry of Housing, Communities and Local Government (2018) [4] in their reports on homelessness stress the need for an integrated approach to homelessness, whilst the United Nations (2020) [2] argue for a multi sectoral approach; this research supports this. Furthermore, this study has highlighted clear and significant gender differences between the experience of being homeless and indeed the disparity in the needs of men and women living on the streets. It is evident that health and social care agencies need to work in partnership with the third sector and social care agencies, as well as working with individuals from the homeless community, if we are to address the structural, cultural and personal barriers that many homeless people face.

### Strengths and Limitations

This research was limited to a defined, local geographical area; as such, the degree to which the findings are generalisable across the UK is limited. That being said, the research provided a wealth of information regarding the experiences of people who are homeless, which are no doubt transferable to other settings. There are, however, limitations to the research, which include not asking length of time of being homeless in the quantitative sample. It was clear from the qualitative arm of this study that the length of time of being homeless is a factor in how individuals perceive and experience homelessness, and this is an area that requires future research. Furthermore, the use of self-reporting questionnaires does, by its nature, rely on transparency, and this is not always ideal, particularly in a population where trust issues are pertinent.

Furthermore, interviews were relatively short, approximately 15–30 min for 1:1 sessions and 50–60 min for the focus groups. This was due to difficulties in participant concentration and focus, which were compounded by COVID-19 restrictions, which meant that they occurred outside. However, due to the development of trust between researcher and participants, a wealth of information was shared.

This study utilised a questionnaire designed by a group of local practitioners including a GP, mental health practitioners, academics and third sector organisations, all of whom had expertise in working with people who were homeless, including an individual with personal experience of homelessness. Although this made for a well-informed set of questions, future research utilising validated self-perceived health questionnaires is warranted to develop a greater understanding of the health and wellbeing issues faced by members of this community. However, as outlined previously, there was a need to ensure that the survey was of an appropriate length to avoid questionnaire fatigue.

## 5. Conclusions

Homelessness is a growing problem and addressing and ameliorating its associated difficulties is a complex one. This study found numerous barriers compounding the problem, from systemic failings and inadequate information sharing to personal difficulties and a perception of prejudice that make seeking help difficult. Furthermore, the experience of being homeless differs according to gender, which further complicates the situation. As a result of these barriers, services, many of which are viewed by the homeless as excellent, are simply not being accessed. Information is poor, hard to find and rarely updated. A reliance on word of mouth to find amenities and help is neither completely effective nor acceptable. It does seem rather unfortunate that the vast majority of the participants felt that it was not a lack of available support that was the issue but a lack of signposting for that support. Although not without their own challenges, technological solutions seem a prudent intervention and a potential future direction for research.

## Figures and Tables

**Figure 1 ijerph-19-00046-f001:**
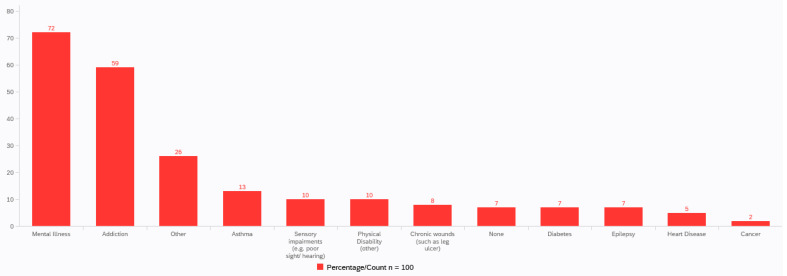
Range of health issues.

**Figure 2 ijerph-19-00046-f002:**
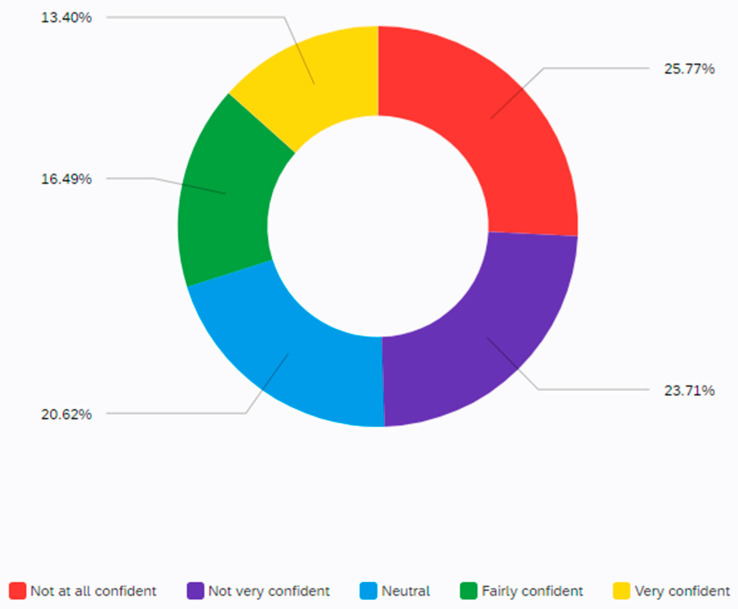
Confidence using the internet (97/100 responded).

**Table 1 ijerph-19-00046-t001:** Categorisations of homelessness (FEANTSA 2018) [3].

People living rough—such as on the streets or public spaces People living in in emergency accommodation—such as overnight shelters People living in accommodation for the homeless—such as homeless hostels, temporary accommodation, transitional supported accommodation, women’s shelter or refuge People living in institution—such as staying longer than needed in hospital due to lack of housing and people in prison with no identified housing prior to release People living in non-conventional dwellings due to lack of housing—such as mobile homes, non-conventional buildings or temporary structures People living temporarily with family and friends due to lack of housing

**Table 2 ijerph-19-00046-t002:** Participant demographics.

Gender	Total	Mean Age	Age Range	Currently Homeless	Previous Homeless
All	100	41.58 (11.72)	20–71	32% (32)	68% (68)
Male	71	41.46 (11.13)	20–66	30% (22)	70% (49)
Female	29	41.50 (14.12)	22–71	34% (10)	66% (19)

**Table 3 ijerph-19-00046-t003:** Participant demographic information.

Code	Method	Gender	Age	Accommodation	Length of Time Homeless	Health Issues
P1	FG1	Male	45	‘Dry’ Charity Home	4 months	No
P2	FG1	Male	43	‘Dry’ Charity Home	18–20 years	Gastric Issues, Addiction
P3	FG1	Female	36	‘Dry’ Charity Home	6 months	Anxiety, Depression
P4	FG2	Male	54	Hotel (LA Funded)	4 months	None
P5	FG2	Male	35	Hotel (LA Funded)	3 months	Depression, Anxiety
P6	FG2	Male	48	Hotel (LA Funded)	6 years	Mental Health
P7	Interview	Male	39	Street	10 Years	Addiction
P8	Interview	Male	47	Night Shelter	3 Months	Mental Health
P9	Interview	Male	63	YMCA	4 Years	Dietary
P10	Interview	Female	50	B and B	4 Years	Physical Health
P11	Interview	Male	66	Sheltered Housing	6 Years	None
P12	Interview	Male	30	Sofa surfing	5 years	None
P13	Interview	Male	49	Hostel	2 Years	None
P14	Interview	Female	22	Street	4 weeks	Mental Health
P15	Interview	Male	49	Street	2 Years	Mental/Physical Health
P16	Interview	Female	39	Streets	2 years	Mental Health

**Table 4 ijerph-19-00046-t004:** Interview guide.

We are interested in hearing about your experiences of accessing and using health services such as doctors, nurses, mental health services, etc., whilst living on the street, can you tell us more about this? ◦ How do you find out where to access the right services for your needs? ◦ Can you tell us about some of the barriers for you in accessing services? We are also interested in hearing about your experiences in accessing and using wider services such as getting food, financial support, keeping yourself dry, find shelter, etc., can you tell us about this? ◦ How do you find out about this information locally? ◦ Can you tell us about some of the barriers for you in accessing these services? How comfortable are you in using technology such as a mobile phone and the internet? ◦ What are the major challenges for you in using a mobile phone and accessing the internet? We are interested in developing a technological application to address some of these issues. What kind of technological solution would be most useful to those living in the street? ◦ How do you think this would help? ◦ What information would be most valuable for you to be able to access and use health and social care services? Is there anything else you wish to add?

## Data Availability

The data is currently being adapted for inclusion into Bournemouth University BORDaR.

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
