# Peer review of "How Do People Who Are Homeless Find Out about Local Health and Social Care Services: A Mixed Method Study"

_ijerph, 2021, doi:10.3390/ijerph19010046_

Round 1
Reviewer 1 Report
Thank you for studying the implications of the social determinants of health in the vulnerable homeless population. I have read and appreciate your research and outcomes.
- The article is clearly written and addresses a pertinent topic, although the outcomes/findings are not new information.
- The introduction sufficiently addresses the background and includes adequate references.
- The research design is appropriate. However, the results are not reproducible given the information in the methods section. Specifically, there is little information on the number of participants in the focus groups versus the one-one interviews. Also not clear if the same individuals participated in both, or only one of these methods. Is the entire number for qualitative 16? This area could use more explanation to more fully describe the qualitative data.
- Also need a much more in depth section on strengths and limitations. There were limitations in this study that were not addressed. Conclusions are appropriate.
- There are some unclear sentences/ missing words in the article, and these should also be addressed. Specifically line 29, 51, 63, and 243. Lines 204-205 and 553 are confusing.
I appreciate the authors' efforts in measuring and reporting on this very important topic for this very vulnerable population. Some additional explanations will benefit the overall article and enhance its readability.
Author Response
Thank you for your suggestions to further strengthen the paper. In order to make transparent corrections made, we have listed them in the table attached, the reviewers’ comments (left hand column) and how we have addressed these in the revised paper (right hand column). In addition, within the paper we have highlighted the relevant sections in yellow in order to make it clear which changes have been made. We hope that these revisions sufficiently address the reviewers’ comments and the paper is acceptable for publication within the International Journal of Environmental Research and Public Health.
Kind Regards

Reviewer 2 Report
In general, congratulate the authors because the relevance of the chosen topic for public health. Also, it is a very important issue that involves significant difficulties in accessing to the target population.
Background
- It is considered necessary to include a more exhaustive review of studies on homeless in the UK and in other countries in the European environment (there are studies in Greece and Spain) and internationally (there are relevant studies in the US) to give greater consistency to the article and the study
-It is considered relevant include a better description of the structure of local health and social care services: number, distribution, type of support offered and form of access to them.
-There is a mistake in the appointment line 46: repeat appointment 1- correct
Aims
-Locate it before the method section, at the end of the introduction. Methods
-Study sample section: Clarify in this section the number of people who fill in the questionnaire. Present more clearly the number of groups and individual interviews, group composition and participant profiles. Focus group segmentation criteria. Criteria to stop sampling.
-Table 3 and 4 is part of the description of these participants
.- Quantitative data collection: attach the questionnaire used. Present data on the validation of the scales used. Present references of studies where this questionnaire has been used. Include all information that helps to establish the validity of the instrument for collecting information through this questionnaire.
-Clarify what “research ream (SR)” refers - line 121 and VH - line 135
- The average duration of the in-depth interviews is 15 minutes. It is a very short time for an interview. Reflect on the discussion about the potential reasons and possible limitations.
Results
-Describe in the two figures the number of people who answered each topic (is it 100% in all cases?). It is suggested to use a table to present all the main quantitative results and according to sex. It can be included as supplementary material.
-Presentation of qualitative data: use self-explanatory results presentation epigraphs (ei Line 280 Theme 1: Obstacles: “perceived obstacles to access to ¿services? Or information about services ?. The same occurs in other epigraphs. Ei. line 281 uses the heading “Structural”
-Use headings that are themselves self-explanatory.
-Delve into theme 3 since it is one of the most relevant in the study.
-Use a table or diagram to present a synthesis of the most relevant analysis categories and topics for each category, even according to structural profiles of different participants (according to sex or other segmentation variables).
-Deepen the qualitative analysis according to population segments (the issues that emerge for people who have been homeless for 4 months are the same as for people who have been homeless for more than 20 years? Is it the same for men and women? The same for people of 20 years or more than 60. What are the most relevant topics according to those profiles.
Discussion-
-Discuss the results in light of the scientific evidence that appears in other studies in similar populations.
Reference
- There are important mistakes in the presentation of the references. Ei. line 595 does not include the year. Line 602 correct the mistake. And, in general, review the citation system according to the rules of this journal
Author Response
We wish to thank the reviewers for their suggestions to further strengthen the paper. In order to make transparent corrections made, we have listed them in the table attached, the reviewers’ comments (left hand column) and how we have addressed these in the revised paper (right hand column). In addition, within the paper we have highlighted the relevant sections in yellow in order to make it clear which changes have been made. Particularly helpful was your suggestion regarding data analysis. As a result we have a much richer findings, in particular relating to time spent homeless, gender and age. This has helped illustrate the different experiences of homelessness. We hope that these revisions sufficiently address the reviewers’ comments and the paper is acceptable for publication within the International Journal of Environmental Research and Public Health.
Kind Regards

Round 2
Reviewer 2 Report
Dear authors
The paper has improved and incorporated most of the suggestions. However, it still has areas for improvement, mainly in terms of methods and reference and the formal aspect of the papers.
I suggest the following changes
1-Answering my question about the questionnaire, the authors say that "present data on the validation of the scales used, and attaching the questionnaire used" is not relevant because the questionnaire has been specifically designed for this research. In this case, it is especially important to test the validity of the questionnaire. It is important to demonstrate that the measurement methods are valid enough to be able to draw conclusions from the study. I suggest attaching in a complementary file the process carried out to design and validate the questionnaire, as well as the questionnaire used. The authors show results on the prevalence of different health measures in this population. Example, percentage of people with self-perceived poor health. There are validated instruments to measure perceived health. Have you used those instruments? . Why haven't you used them? what is the instrument used in this research. Why ?. This are some questions that I could ask and that could be answered by improving the description of the methodology. 2-In relation of the duration of the in-depth interviews I suggest include in the discussion section the explanation of the authors. Also, in my opinion, this circumstances makes qualitative methods especially suitable for collecting information in these contexts. I am sure that there are previous studies that could be cited in this sense than you could add to this point. 3- Figure 2 are the numbers around the circle percentages? Please correct them. 4- Line 513: What does “patinant” mean?5- Line 521 : what does Y(MCA)´ mean?6- Review all the documents there are several spelling and grammar errors (ej. Line 545 memebrs) 7- Textual citations use different formats: sometimes italics are used and sometimes not, sometimes double spacing is used and sometimes not, sometimes the profile of the respondent is described and sometimes not. Please select a structure for the citations and use it in all cases 8- Reference: There are mistakes in references. please check the journal's guidelines regarding bibliographic references and use them. Also reference of line 787 (line 602 in previous version) has been not corrected.
Author Response
Good afternoon,
Thanks for your input with this paper. Please find attached a table with all the issued for consideration now addressed. The changes are highlighted in the paper for ease of finding.
